# Usefulness of ^18^F-FDG PET-CT in the Management of Febrile Neutropenia: A Retrospective Cohort from a Tertiary University Hospital and a Systematic Review

**DOI:** 10.3390/microorganisms12020307

**Published:** 2024-01-31

**Authors:** Andrea Gutiérrez-Villanueva, Claudia Quintana-Reyes, Elena Martínez de Antonio, Begoña Rodríguez-Alfonso, Karina Velásquez, Almudena de la Iglesia, Guiomar Bautista, Cristina Escudero-Gómez, Rafael Duarte, Ana Fernández-Cruz

**Affiliations:** 1Infectious Diseases Unit, Internal Medicine Department, Hospital Universitario Puerta de Hierro-Majadahonda, Instituto de Investigación Sanitaria Puerta de Hierro-Segovia de Arana, 28222 Madrid, Spain; agutierrezv@salud.madrid.org; 2Hematology Department, Hospital Universitario Puerta de Hierro-Majadahonda, 28222 Madrid, Spain; cquintanar@salud.madrid.org (C.Q.-R.); emdeantonio@salud.madrid.org (E.M.d.A.); mariaalmudena.de@salud.madrid.org (A.d.l.I.); guiomar.bautista@salud.madrid.org (G.B.); rafael.duarte@salud.madrid.org (R.D.); 3Nuclear Medicine Department, Hospital Universitario Puerta de Hierro-Majadahonda, 28222 Madrid, Spain; brodrigueza@salud.madrid.org (B.R.-A.); karinaelisabet.velasquez@salud.madrid.org (K.V.); 4Medical Library, Hospital Universitario Puerta de Hierro-Majadahonda, 28222 Madrid, Spain; cescuderog@salud.madrid.org; 5Facultad de Medicina, Universidad Autónoma de Madrid (UAM), 28049 Madrid, Spain

**Keywords:** PET, febrile neutropenia, ^18^FDG-PET-CT, review

## Abstract

Febrile neutropenia (FN) is a complication of hematologic malignancy therapy. An early diagnosis would allow optimization of antimicrobials. The ^18^F-FDG-PET-CT may be useful; however, its role is not well established. We analyzed retrospectively patients with hematological malignancies who underwent ^18^F-FDG-PET-CT as part of FN management in our university hospital and compared with conventional imaging. In addition, we performed a systematic review of the literature assessing the usefulness of ^18^F-FDG-PET-CT in FN. A total of 24 cases of FN underwent ^18^F-FDG-PET-CT. In addition, 92% had conventional CT. In 5/24 episodes (21%), the fever was of infectious etiology: two were bacterial, two were fungal, and one was parasitic. When compared with conventional imaging, ^18^F-FDG-PET-CT had an added value in 20 cases (83%): it diagnosed a new site of infection in 4 patients (17%), excluded infection in 16 (67%), and helped modify antimicrobials in 16 (67%). Antimicrobials could be discontinued in 10 (41.6%). We identified seven publications of low quality and one randomized trial. Our results support those of the literature. The available data suggest that ^18^F-FDG-PET-CT is useful in the management of FN, especially to diagnose fungal infections and rationalize antimicrobials. This review points out the low level of evidence and indicates the gaps in knowledge.

## 1. Introduction

Febrile neutropenia (FN) is a common complication of cancer, particularly in patients with hematological malignancies who receive high-intensity chemotherapy or stem cell transplantation (SCT) [1]. In this setting, infections are an important complication and cause significant morbidity and mortality. The etiological workup of FN includes, in addition to microbiological tests, different imaging tests; among them, the most commonly used is computed tomography (CT). Specifically, high-resolution computed tomography is recommended for suspected invasive fungal infection (IFI) [2].

Unfortunately, in a large number of cases it is not possible to determine the cause of the fever with conventional imaging. This results in the prolonged use of empirical broad-spectrum antimicrobials, both antibiotics and antifungals [3]. In addition, in almost 50% of the cases according to some studies [4], the etiology of FN is not infectious. In these situations, an early etiological diagnosis would allow the performing of targeted diagnostic tests and optimization of antimicrobial therapy, including timely discontinuation of unnecessary antimicrobials [4]. Therefore, there is a need to implement better complementary tests to improve diagnosis and allow de-escalation of antimicrobials.

PET with ^18^F-fluorodeoxyglucose provides functional information that correlates with the anatomical data provided by CT. Using ^18^F-FDG as a radiotracer, information on the metabolism in different tissues is obtained. Unlike conventional CT, ^18^F-FDG-PET-CT is capable of evaluating more than one area of the body in just one session in addition to providing metabolic information, allowing one to more easily detect clinically silent lesions [5]. This imaging technique is typically used for staging cancer and has seen its use increase lately as part of the study of fever of unknown origin or FN [1,6].

In recent years, several studies have shown the potential of ^18^F-FDG PET-CT to localize the source of fever and to differentiate between infection and other etiologies in patients with FN. Although ^18^F-FDG-PET-CT has not shown a clear benefit in the differential diagnosis of cancer and infection, nevertheless, it could have a role in FN and in detecting the spread of infection and occult infection [7]. Other authors have underlined the high negative predictive value of ^18^F-FDG-PET-CT that facilitates the adjustment of antimicrobial treatment [4,8]. In particular, they point out that ^18^F-FDG-PET-CT could play an important role in the diagnosis of IFI and help with the withdrawal of empirically initiated antifungals [1,4,8]. In a recent review on the management of patients with high-risk FN, the authors suggest that ^18^F-FDG-PET-CT could be especially helpful in these patients when it comes to reducing the antibiotic spectrum without changes in ICU admissions or mortality [9].

However, the literature on this topic is scarce and mainly consists of short cases series in diverse settings. In addition, most of these studies do not compare the performance of conventional tests and ^18^F-FDG-PET and the added value provided specifically by ^18^F-FDG-PET-CT. Consequently, its role in routine clinical practice has not been well established so far [1,4,6,8].

Although several authors have reviewed the existing literature [1,4,6,7,8], to our knowledge, there is not a comprehensive and systematic review on this topic.

In the present article, we analyze our center’s data on the usefulness of ^18^F-FDG PET-CT in hematological patients with FN and its added value compared with conventional imaging, and we perform a systematic review of the published literature on the use of ^18^F-FDG PET-CT in that setting.

## 2. Patients and Methods

### 2.1. Single-Center Retrospective Cohort Study

#### 2.1.1. Design, Study Period, and Subjects

Our institution is a 613-bed tertiary-care teaching hospital in Madrid, Spain. The Hematology Department has an active SCT program, including allogeneic SCT (haploidentical and cord as well).

We performed a retrospective observational study including all adult patients admitted to Puerta de Hierro-Majadahonda Hospital between 2015 and 2022 diagnosed with hematological malignancies (leukemia, aplasia, myelodysplastic syndrome, multiple myeloma, or lymphoma) undergoing chemotherapy or SCT who underwent at least one ^18^F-FDG PET-CT as part of the FN management.

#### 2.1.2. Data Collection

Epidemiological, clinical (including the type of hematological malignancy and SCT, the type of infection, localized or disseminated disease, and the type of pathogen), laboratory, and imaging data were extracted from electronic medical records (SELENE System, Cerner Iberia, S.L.U., Madrid, Spain) using a standardized data collection form. The ^18^F-FDG PET-CT indication and impact of the results on FN management were specifically addressed. All data were included by a primary reviewer and, subsequently, checked by two senior physicians.

#### 2.1.3. ^18^F-FDG PET-CT Technique

All ^18^F-FDG PET-CT scans were performed according to EANM (European Association of Nuclear Medicine) guidelines, in hybrid PET/CT chamber systems [10]. The CT component was non-contrast enhanced. All patients complied with a previous fasting period of at least six hours (12–18 h in cases of suspected endocarditis; in this case a dietary modification protocol was also applied). Ideally, they should maintain blood glucose levels lower than 180 mg/dL. If insulin was administered, the injection of ^18^F-FDG would be spaced at least four hours apart. For infectious and inflammatory diseases, the same acquisition, reconstruction, and post-processing described in the procedures of the EANM for tumors were used [10,11]. Full-body ^18^F-FDG PET-CT, from the cranial vertex to the feet in a supine position, was acquired approximately 50–60 min after the intravenous injection of 370 ± 30 MBq ^18^F-FDG depending on the patient’s weight. When infective endocarditis was a possibility, the study was completed with dedicated cardiac ^18^F-FDG PET-CT acquisition.

The ^18^F-FDG PET-CT images were analyzed for increased uptake of ^18^F-FDG outside the areas of physiological incorporation. A qualitative analysis was carried out, considering the uptake pattern (focal, linear, or diffuse) and the distribution of the radiotracer in the pathological area or lesion (homogeneous or heterogeneous), and a semiquantitative analysis was performed considering the intensity of the uptake. The images were interpreted as normal, equivocal, or with pathological uptake according to the standard uptake values (SUV): the visual scores were 0, no pathological uptake; 1, uptake similar to the vascular pool in the mediastinum; 2, uptake higher than the vascular pool but lower than the liver pool; 3, uptake similar to or slightly higher than the liver; 4, uptake clearly higher than the hepatic, where 0 and 1 would be negative and 2, 3, and 4 would be positive (always assessing the location and alternative causes that explain the uptake).

#### 2.1.4. Other Imaging Techniques

The diagnostic workup for FN was performed at the discretion of the treating physicians. For every case, the results of conventional imaging techniques performed during the episode were compared with the ^18^F-FDG PET-CT results, according to the reports by radiology specialists (or cardiologists, when applicable). This included X-ray, CT, MRI, and, in the case of bloodstream infection caused by Gram positives or yeasts, echocardiography.

#### 2.1.5. Definitions

We followed the criteria for febrile neutropenia as per the NCCN guidelines [12]:-For fever, a single temperature equivalent to ≥38.3 °C orally or equivalent to ≥38.0 °C orally over a 1 h period;-For neutropenia, ≤500 neutrophils/mcL or ≤1000 neutrophils/mcL and a predicted decrease to ≤500/mcL in the next 48 h.

In addition, we also evaluated patients with persistent low-grade fever (temperature > 37.5 °C for more than 72 h).

#### 2.1.6. Usual Care

For antimicrobial prophylaxis, see Appendix A (for acute myeloid leukemia (AML), levofloxacin and posaconazole during the neutropenia period and acyclovir in cases who receive fludarabine; for acute lymphoblastic leukemia (ALL), cotrimoxazole and acyclovir and posaconazole when finishing vincristine; for autologous SCT, cotrimoxazole impregnation prior to transplantation, levofloxacin, fluconazole, and acyclovir; for allogeneic SCT, cotrimoxazole impregnation prior to transplantation followed by nebulized pentamidine, levofloxacin, posaconazole, and acyclovir, plus letermovir and azithromycin in high-risk patients).

Regarding empirical antimicrobials, before 2019, empirical therapy for febrile neutropenia consisted of meropenem. From 2019 on, empirical therapy consisted of piperacillin-tazobactam or cefepime (the latter in cases where no anaerobic coverage was deemed necessary). No surveillance cultures were obtained routinely, but in cases known to be colonized by resistant microorganisms, antimicrobial therapy was adjusted accordingly.

In both periods, teicoplanin was added in cases of catheter infection suspicion, and amikacin was added in cases of septic shock.

#### 2.1.7. Data Analysis

Quantitative variables are expressed as means and standard deviations (SD) and/or medians and interquartile ranges (IQR), and qualitative variables are expressed as frequencies and percentages. Measures of central tendency (mean and SD, and median and IQR) and proportions were calculated with IBM SPSS Statistics 22.

### 2.2. Systematic Literature Review

The studies were identified through a systematic search in different bibliographic databases using search terms (MeSH) related to the topic, specifically, Pubmed, Embase, and Cochrane Library. These databases were searched without language or publication date restrictions (see search strategy in Appendix A). We did not exclude articles based on the retrospective or prospective nature of the study. The reference lists of the relevant studies were checked to identify additional relevant articles.

To be eligible, a study had to evaluate the use of ^18^F-FDG PET-CT in the management of FN. References were screened by two researchers based on the title and abstract using the PICOS framework (Table 1). Irrelevant references were excluded with explicit reasons. In a second step, the remaining references were screened based on the full text.

The following data were obtained for each included study (using a standardized form): the title, reference, study design, source of funding, country and setting, sample size, duration and follow-up, details of the statistical analysis, eligibility and exclusion criteria, patient and disease characteristics, intervention and comparator characteristics, and limitations/comments regarding the study. Two researchers performed the data extraction, which was checked by two senior researchers. Reporting was conducted in accordance with the PRISMA guidelines for systematic reviews.

The quality assessment of the included studies was carried out by two assessors who independently used the Cochrane Collaboration’s tool for assessing the risk of bias [13]. Discrepancies in the scores were resolved through discussion.

This systematic review was performed according to the recommendations of the PRISMA guidelines. A checklist is included as Appendix A.

## 3. Results

### 3.1. Single-Center Retrospective Cohort Study

Among 638 eligible patients with FN during the study period, 24 episodes of FN were detected in 23 patients who underwent ^18^F-FDG PET-CT as part of the FN workup (Figure 1).

#### 3.1.1. Characteristics of the Patients with FN That Underwent an ^18^F-FDG PET-CT

The characteristics of the patients who underwent ^18^F-FDG PET-CT for FN study are displayed in Table 2. Fifty-seven percent were men. The mean age was 58.6 years (SD 19.6). Regarding the non-onco-hematological comorbidities, 17% had diabetes, 8% had at least moderate chronic kidney disease, and 4% had chronic pulmonary disease (COPD). The most frequent onco-hematological underlying disease was acute myeloid leukemia in 12 patients (50%), followed by acute lymphoblastic leukemia (ALL) in 3 patients (12%), multiple myeloma (MM) and myelodysplastic syndrome (MDS) in 2 patient each (8%), and NK immunodeficiency in 1 patient (4%). Four patients (17%) were stem cell transplantation recipients. The reasons for the SCT were as follows: ALL in two patients, MM in one patient, and MDS in one patient.

The majority of patients fulfilled criteria for persistent fever (58%), followed by patients with persistent low-grade fever (33%). Neutrophil counts on the day of the ^18^F-FDG PET-CT were below 100 cells/mm^3^ in 12 (50%) and between 100 and 500 cells/mm^3^ in 10 (42%) patients. Two (8%) had neutrophil counts above 500 cells/mm^3^ on the day the test was performed but had recently recovered from severe neutropenia during the previous week.

Patients were receiving antimicrobial prophylaxis according to risk factors, following the institution’s protocols (Appendix A). The distribution of the empirical treatments administered at the moment of ^18^F-FDG PET-CT is shown in Table 1, the most commonly used drugs being meropenem (54%) and piperacillin-tazobactam (37%).

#### 3.1.2. Fever Etiology (Table 2)

In five cases of 24 FN episodes (21%), fever was considered of infectious etiology. The etiology was bacterial infection in two cases (8%), fungal in two (8%), and parasitic in one (4%). No viral infection was diagnosed. In one case (4%) there was a clinical diagnosis of infection, but a microbiological etiology could not be determined (catheter uptake without isolation of microorganisms). Among the bacterial infections, two bloodstream infections were diagnosed: a case of catheter-related *S. haemolyticus* bacteremia and another of catheter-related persistent *E. faecium* bacteremia without catheter uptake or septic metastases. Regarding fungal infections, one patient presented an invasive fusariosis and another with possible pulmonary IFI. The remaining patient with a known infectious etiology had visceral leishmaniasis.

In all but one patient, the infection was localized. The disseminated case was a patient initially diagnosed of naso-sinusal fusariosis, where the initial ^18^F-FDG PET-CT helped to diagnose the occult source of fever, and in addition, the monitoring ^18^F-FDG PET-CT detected the dissemination of the infection.

The median duration of antimicrobial therapy until ^18^F-FDG PET-CT was performed was 13.5 days (3–30). The fever was considered to be secondary to non-infectious causes in 20 cases (83.3%): in 11 (46%) it was secondary to the underlying hematologic malignancy, in 1 (4%) it was considered of inflammatory etiology (reduction in corticosteroids in the context of disseminated mycobacterial infection), there were 3 cases (12%) of engraftment syndrome and 2 of graft-versus-host disease (GVHD) (8%), and in 3 cases (12%) an etiology was not identified.

#### 3.1.3. Characteristics of the ^18^F-FDG PET-CT as Compared with Conventional Imaging

The indication for ^18^F-FDG PET-CT was the study of FN in all patients. The median time of neutropenia before ^18^F-FDG PET-CT was 13.5 days (3–74), and the median time to ^18^F-FDG PET-CT from the beginning of fever was 13 days (3–28).

In addition, all had undergone conventional imaging before ^18^F-FDG PET-CT. Most of them had undergone CT: 16 patients (67%) had undergone conventional body CT, 4 (17%) had undergone chest CT, 3 (12%) had undergone abdominal CT, and 2 patients (8%) had undergone sinus CT, while 2 (8%) had undergone MRI, 3 (12%) had had a transthoracic echocardiogram, and 2 (8%) had had a transesophageal echocardiogram. In addition, two patients (8%) underwent digestive endoscopy, and another two (8%) underwent bronchoscopy. The median time from conventional imaging to ^18^F-FDG PET-CT was 13 days (1–29).

The ^18^F-FDG PET-CT showed pathological uptake in 20 (83%) cases. The most frequent location of this uptake was intra-abdominal visceral (37%, mainly hepato-splenic uptake), followed by bone marrow and lung uptake (21%). The most common distribution was multifocal in 71% of cases followed by focal uptake in 12%. The remaining 17% did not show any ^18^F-FDG uptake.

Table 3 shows the results of conventional imaging compared with those of ^18^F-FDG PET-CT and the added value of ^18^F-FDG PET-CT in the management of febrile neutropenia. The ^18^F-FDG PET-CT provided added value to the previous conventional imaging study in 20 patients (83%) considering as such the diagnosis of new infection or the exclusion of infection that led to the modification of antimicrobials or the initiation of treatment of the underlying hematological disease. It contributed to the diagnosis of new sites of infection in 4 patients (17%), ruled out infection in 16 patients (67%), and helped modify antimicrobial treatment in 16 patients (67%). It also allowed starting chemotherapy or immunomodulators in four patients: two patients started chemotherapy, one patient received specific treatment for graft-versus-host disease, and corticosteroid doses were increased in another patient.

Pathological uptake in ^18^F-FDG PET-CT helped perform targeted diagnostic tests in 58%: four patients (17%) underwent fine-needle aspiration/biopsy of the pathological uptake area, and one patient (4%) underwent bronchoscopy, among other complementary tests.

The results of ^18^F-FDG PET-CT implied the removal of the venous catheter in one case and surgical debridement in another one for source control.

In the cases that were eventually diagnosed as infection, the most common site of infection identified by ^18^F-FDG PET-CT was hepatosplenic and biliary (8%), followed by catheter and pulmonary (4%).

Among the non-infectious causes, the most common reason for pathological uptake in ^18^F-FDG PET-CT was the underlying disease in 11 patients (46%).

Concerning antimicrobial therapy, in 16 patients (67%) the antimicrobial spectrum was modified based on ^18^F-FDG PET-CT results. In 2 (8%) patients, antimicrobials were de-escalated; in 1 (4%) case the spectrum was expanded; and in 10 (42%) patients it allowed the discontinuation of antimicrobials. There was a need to start new antimicrobial treatment in three (12%) patients, in one of the cases also accompanied by surgical debridement.

Only one patient had more than one ^18^F-FDG PET-CT during the study of the episode of FN. The patient diagnosed with sinus IFI due to Fusarium had a control ^18^F-FDG PET-CT scan performed one month after the first one, which, as aforementioned, detected dissemination of the infection to the lungs as well as persistence of the sinusal infection.

### 3.2. Systematic Literature Review

#### Search Strategy and Inclusion

The literature search retrieved 341 references that were de-duplicated, and non-English, Spanish, and French references were excluded (Figure 2). The remaining references were screened for eligibility based on the titles and abstracts (of which 160 were excluded). Based on full-text evaluation of the remaining publications, 16 articles that evaluated the use of ^18^F-FDG PET-CT in the management of FN were included. This resulted in a sample of one RCT and 15 publications. Among them, five narrative reviews and a survey were excluded from the present analysis. Two-case reports were excluded to avoid publication bias. Eight articles were selected (Table 4).
Quality appraisal

The quality of the eight included publications was evaluated as moderate to poor. The methodology used was heterogeneous. Because of the nature of the interventions, blinding of the patients and staff was not possible. Three of the studies were retrospective and, thus, non-random by design. Among the prospective studies, in one case, there was no comparison of ^18^F-FDG PET-CT and conventional techniques, whereas the remaining studies performed both techniques on the same patients sequentially; therefore, there was no randomization to one or the other study. Likewise, there was no randomization of the sequence in which the techniques were performed.

Only one open-label randomized controlled trial was identified. Although masking of the randomization was not possible, the clinical impact of the randomized scans and the cause of neutropenic fever were assessed by an independent adjudicating committee to reduce the risk of bias.
Results of the studies according to the methodology (Figure 2)


Clinical trial


Recently, a multicenter phase 3 controlled clinical trial [3] was published that randomized patients with high-risk NF 1:1 to perform CT vs. PET-CT. The primary endpoint was a combination of starting, stopping, or changing the spectrum of antimicrobial therapy as a result of the information provided by the imaging technique. They included a total of 134 patients (PET-CT 65; CT 69). Antimicrobial rationalization occurred in 82% of patients in the ^18^F-FDG PET-CT group and 65% in the CT group. The most frequent action was the reduction in the spectrum of antimicrobial therapy, 43% for ^18^F-FDG PET-CT compared with 25% for CT (*p* = 0.024). The authors concluded that ^18^F-FDG PET-CT was associated with better optimization of antimicrobial therapy and could help decision making in this type of patient. The drawback of this clinical trial was the lack of direct comparison of ^18^F-FDG-PET-CT and conventional imaging in the same patient. The authors did not provide information about whether the differences in baseline characteristics or the final fever etiology between patients who underwent ^18^F-FDG PET-CT or conventional CT were statistically significant.
2.Original articles

Seven original articles that studied the usefulness of ^18^F-FDG PET-CT in FN were found through the systematic search, five of them prospective and two retrospective [14,15,16,17,18,19,20]. The characteristics of the original articles that evaluated ^18^F-FDG-PET-CT’s usefulness for FN management are summarized in Table 3.
Studies comparing the results of conventional tests and ^18^F-FDG PET-CT in the same patient

Four articles [14,15,19,20] compared conventional imaging and ^18^F-FDG-PET-CT performed in the same patient as part of the study of FN, in order to assess which one provides more information to improve management. Only one of these provided individual data with a head to head comparison of conventional imaging with ^18^F-FDG-PET-CT in the same patients [19]. A total of 161 patients with FN were evaluated in these studies (147 adults and 14 children). The median time to ^18^F-FDG PET-CT from the beginning of fever to the performance of ^18^F-FDG PET-CT varied between 6 and 14 days. The median time from conventional imaging to ^18^F-FDG PET-CT was not provided in any of these studies. The final diagnosis was infectious in a high proportion of cases, varying from 55% to 79%.

Camus et al. [14] carried out a prospective single-center study to investigate the ability of ^18^F-FDG-PET-CT to find the source of infection in patients with FN. In this study, among the 38 patients with a final clinical diagnosis of infection (79%), 23 had a pathological FDG uptake, resulting in a ^18^F-FDG-PET-CT sensitivity of 61%. Among the 17 patients diagnosed with pneumonia by conventional evaluation, ^18^F-FDG-PET-CT detected pulmonary uptake in 11 (64.7%) and uptake at multiple levels in 6 (35.3%). Gafter-Gvili et al. also evaluated in a prospective study the performance of ^18^F-FDG-PET-CT for the diagnosis and treatment of infections in high-risk patients with FN [15]. In this case, the sensitivity of ^18^F-FDG-PET-CT was 79.8% compared with 51.7% for chest/sinus CT alone. The specificities were 32.14% versus 42.85%. Furthermore, in more than 50% of patients, ^18^F-FDG-PET-CT changed the pre-test diagnosis and helped modify patient management. Both studies concluded that ^18^F-FDG PET-CT had the ability to assist in the evaluation and management of these patients.

**Table 4 microorganisms-12-00307-t004:** Summary of the eight articles selected by means of the systematic search.

Authors	Type of Study	Study Population	FN	Compare CT + ^18^F-FDG-PET-CT	Relevant Results	Final Diagnosis	Limitations
Mahfouz T et al. [18] Arkansas (USA)	Retrospective study, 2005	Multiple myeloma. (A total of 165 infectious episodes were identified in 143 patients with MM; 27 episodes of neutropenia.)	No	No	^18^F-FDG-PET-CT in patients with MM identified lesions not detectable by other methods on 46 occasions, determined disseminated infection on 32, helped modify therapies in 55 episodes, and detected 20 clinically relevant silent infections. Guided diagnostic tests: No specific data.	Does not provide specific data of the neutropenic patients	Retrospective. Single center. Myeloma only. No separate data of the 27 cases of neutropenia. Does not compare CT vs. ^18^F-FDG-PET-CT.
Koh KC et al. [17] Australia	Retrosective study (case-control), 2012	Hematologic malignancies. (100% patients with FN.) Median time from CT compared with ^18^F-FDG-PET-CT: 4.2 d.	Yes	Yes (in different patients)	CT (*n* 76) vs. ^18^F-FDG-PET-CT (*n* 37) in FN of unknown origin. An underlying cause for FN was determined in 94.6% of cases (^18^F-FDG-PET-CT), compared with 69.7% of controls (CT). ^18^F-FDG-PET-CT had a significant impact on antimicrobial utilization compared with conventional imaging (35.1% vs. 11.8%). Guided diagnostic tests: No specific data.	Infection: 67.6% in cases vs. 67.1% control group. IFI: four cases.	Single center. Does not compare CT and ^18^F-FDG-PET-CT scans in the same patient.
Vos FJ et al. [16] Netherlands	Prospective cohort study, 2012	Hematologic malignancies (AML; MDS) and HSCT (100% with neutropenia, 26 of 28 FN). Mean time from starting chemotherapy to ^18^F-FDG-PET-CT: 14 d.	Yes	No	^18^F-FDG-PET-CT scans were performed on patients with NF and elevated CRP > 50 mg/L. *n* = 28. Pathological findings were found in 26 cases (18 gastrointestinal, 9 related to CVC, and 7 related to lung). Guided diagnostic tests: yes (ultrasound in case uptake in the CVC tract).	26/28 FDG uptake of infectious origin. IFI: seven cases.	Single center. Does not compare the performance and findings of CT and ^18^F-FDG-PET-CT scans in the same patient.
Camus V et al. [14] France	Prospective study, 2015	Hematologic malignancies (AML; ALL) and HSCT (100% FN). Median days of neutropenia: 15. Median days of fever: 14.	Yes	Yes (in the same patient)	Usefulness of ^18^F-FDG-PET-CT in detecting the source of infection in FN. *n* = 48. In 31 cases, there was a pathological uptake. In 13, diagnosis of multiple foci/dissemination. Guided diagnostic tests: only in some patients.	Infection: 79%. IFI: three aspergillosis.	Single center. Few patients. They perform CT and ^18^F-FDG-PET-CT scans in the same patient, but it does not compare them with each other.
Guy SD et al. [19] Australia	Prospective study, 2012	Hematological and solid malignancies. (100% FN.) Median days of neutropenia: 9. Median days of fever: 5–7.	Yes	Yes (in the same patient)	Patients with NF who undergo a ^18^F-FDG-PET-CT scan in addition to conventional techniques. *n* = 20. ^18^F-FDG-PET-CT identified nine infections that CT did not and had a clinical impact in 75% of patients. Compares ^18^F-FDG-PET-CT and conventional imaging in the same patient and provides individual patient data. Guided diagnostic tests: only in some patients.	Infection: 11/20 patients (55%)	Single center. Few patients. Does not include allogeneic HSCT.
Gafter-Gvili A et al. [15] Israel	Prospective study, 2013	Hematologic malignancies (AML, ALL, lymphoma) and HSCT (100% FN). Median days of neutropenia: 11. Median days of fever: 6.	Yes	Yes (in the same patient)	Use of ^18^F-FDG-PET-CT in high-risk NF vs. conventional techniques, focused on IFI. *n* = 79. ^18^F-FDG-PET-CT changed diagnosis in 69% of patients and management in 55%. Guided diagnostic tests: yes.	^18^F-FDG-PET-CT is useful for diagnosis in NF. Infection: 89/117 diagnoses, mainly intra-abdominal infections. IFI: 27 infections.	Single center. Focused on IFI although it also detects other sources of infection.
Wang SS et al. [20]	Retrospective study, 2017	Hematologic malignancies, HSCT and solid malignancies (100% FN)	Yes	Yes (in the same patient)	To assess the impact of ^18^F-FDG-PET-CT on persistent or recurrent fever. *n* = 14. In 11 of them (79%), ^18^F-FDG-PET-CT had a clinical impact: in three, treatment was de-escalated, and in five, antibiotics were discontinued. ^18^F-FDG-PET-CT scans identified new foci in seven patients. ^18^F-FDG-PET-CT helped the final diagnosis in 6 out of 10 patients who had a known cause of fever. Guided diagnostic tests: No specific data.	Infection: 8/14 patients (57.1%). IFI: three cases.	Single center. Retrospective. Pediatric patients only. Few patients. Long time until the ^18^F-FDG-PET-CT scan is performed.
Douglas A et al. [3] Australia	Phase 3 randomized 1:1 multicenter clinical trial of CT vs. ^18^F-FDG-PET-CT/CT, 2022	Hematologic malignancies and HSCT. (100% FN.) Median days of neutropenia: 12. Median days of fever: 8. Median time from CT compared with ^18^F-FDG-PET-CT: 5.5 h.	Yes	Yes (in different patients)	Total *n* = 65 patients in the ^18^F-FDG-PET-CT group and 69 in the CT group. Antibiotic adjustment occurred 82% in ^18^F-FDG-PET-CT and 65% in CT, most frequently reducing the spectrum of therapy, in 28 (43%) of 65 patients in the FDG-^18^F-FDG-PET-CT -CT group compared with 17 (25%) of 69 patients in the CT group. ^18^F-FDG-PET-CT is useful for adjustment of empiric therapy (cessation or reduction in antimicrobials). Guided diagnostic tests: no specific data.	Infection: microbiologically confirmed 72% in cases vs. 57% in controls (CT). IFI: 6% (vs. 4% in controls).	It does not compare the performance and findings of CT and ^18^F-FDG-PET-CT scans in the same patient.

A third prospective study, carried out by Guy et al. [19], which included 20 patients with NF who underwent ^18^F-FDG-PET-CT in addition to conventional techniques, revealed that ^18^F-FDG-PET-CT was able to identify nine infections that CT was not able to identify and had a clinical impact in 75% of patients since it inducedtreatment changes. Like previous articles, it concluded that ^18^F-FDG-PET-CT was useful in the assessment of NF.

A last, retrospective study [20], carried out in a pediatric population that included 14 patients, observed that ^18^F-FDG PET-CT had a positive impact in 11 patients (79%), favoring the rationalization of antimicrobials in three (21%) and their discontinuation in five (36%). Furthermore, compared with conventional tests, it helped identify new sites of infection in seven (50%) patients and contributed to the final diagnosis in six (43%) patients. As in previous articles, the authors consider the potential usefulness of ^18^F-FDG-PET-CT as part of the study of FN.

Another aspect to highlight is the usefulness of ^18^F-FDG-PET-CT to assess the dissemination of infections. In the articles by Camus and Gvili discussed previously, ^18^F-FDG-PET-CT was used to detect occult lesions that led to the diagnosis of disseminated infection in 27% and 1.3% of cases, respectively [14,15].

All of these studies [14,15,19,20], emphasize the usefulness of ^18^F-FDG-PET-CT in the diagnosis of fungal infection and the rationalization of antifungal treatment.
b.Studies that do not compare conventional tests and ^18^F-FDG PET-CT in the same patient

Three articles evaluated the contribution of ^18^F-FDG PET-CT in the diagnosis of infection without doing a head to head comparison with conventional tests in the same patient. These studies either performed conventional tests and ^18^F-FDG PET-CT in different patients [17] or performed only ^18^F-FDG PET-CT [16,18]. A total of 92 patients with FN were evaluated in these studies.

The retrospective study by Koh KC et al. [17] evaluated the impact of ^18^F-FDG-PET-CT on the use of antimicrobials in FN. They identified two groups: one that had undergone ^18^F-FDG-PET-CT (*n* 37) and another that had had conventional imaging (*n* 76). There were no significant differences between cases and controls with respect to age, sex, underlying malignancy, and chemotherapy. The ^18^F-FDG-PET-CT determined the cause of FN in 94.6% of patients compared with 69.7% in the conventional imaging group. Furthermore, ^18^F-FDG PET-CT had a significant impact on antimicrobial use compared with conventional imaging (35.1% vs. 11.8%; *p* 0.003) and was associated with a shorter duration of antifungal therapy. The authors stated that ^18^F-FDG PET-CT improved diagnostic performance and allowed the rationalization of antimicrobials in these patients. In this study as well, the usefulness of ^18^F-FDG-PET-CT for the diagnosis of IFI and the rationalization of antifungal treatment was evidenced.

The two remaining studies did not compare ^18^F-FDG-PET-CT with conventional imaging. The retrospective study by Mahfouz T et al. [18] reviewed the contribution of ^18^F-FDG-PET-CT performed to 248 patients with multiple myeloma (MM) for the staging or diagnosis of infection where there was an uptake atypical for myeloma that could be suggestive of infection. A total of 165 infections were identified in 143 adults with MM, 27 of these episodes being in the context of neutropenia. The ^18^F-FDG PET-CT detected 46 infections not detectable by other methods, helped determine the extent of infection in 32 episodes, and led to modification of the diagnosis and therapy in 55. In patients with staging ^18^F-FDG PET-CT, twenty silent infections were detected. They concluded that ^18^F-FDG PET-CT in MM was a useful technique for diagnosing infection; unfortunately, the authors did not provide specific results for the subset of patients with FN.

The prospective study by Vos FJ et al. [16] included 28 hematological patients with neutropenia who underwent ^18^F-FDG-PET-CT in cases of CRP levels greater than 50 mg/L. In 26 out of 28 (92.9%) patients, that increase in CRP levels was accompanied by fever. The median time from starting chemotherapy to ^18^F-FDG PET-CT was 14 days. They found pathological FDG uptake in 26 of 28 cases (92.9%). The authors did not specify in how many cases ^18^F-FDG-PET-CT guided the performance of diagnostic tests. In this study, pulmonary uptake was significantly associated with the presence of IFI (*p* = 0.04). They determined that ^18^F-FDG-PET-CT in the context of increased CRP was capable of detecting infection in situations of severe neutropenia. An evaluation of the impact of ^18^F-FDG-PET-CT on antimicrobial prescriptions was not provided.

## 4. Discussion

Our data indicate that ^18^F-FDG-PET-CT is useful in the management of FN. In 87% of the cases it helped to confirm or rule out infection, allowing optimization of empirical antimicrobial treatment including de-escalation or discontinuation of unnecessary antimicrobials in 16 cases (67%). These results support data from the 318 total cases with ^18^F-FDG-PET-CT for FN analyzed in the studies included in the present review.

To the best of our knowledge, the present study is one of the few that compares the performance of conventional tests and ^18^F-FDG PET-CT in the same patient during the FN episode. When comparing conventional imaging with ^18^F-FDG-PET-CT performed on different patients, the differences in underlying diseases or in fever etiology could account for the differences observed in the yields of these tests. Comparing both techniques in the same patient overcomes this limitation. In addition, only patients who were still neutropenic at the moment of ^18^F-FDG-PET-CT were selected (with the exception of two patients who had recovered neutrophils very recently, fewer than 3 days before PET), so that we cannot attribute the better performance of ^18^F-FDG-PET-CT to neutrophil recovery.

The ^18^F-FDG PET-CT was especially relevant in the diagnosis of uncommon fungal and parasitic infections, such as fusariosis or leishmaniasis. Interestingly, in the present series the proportion of infectious etiology was low, only 16.7%. Being a retrospective study, we cannot exclude that only patients with a lower probability of infectious cause (non-responders to antimicrobials, with already negative prior tests) underwent ^18^F-FDG-PET-CT. In any case, the ability to rule out infection in these cases is the reason why antimicrobial therapy could be adjusted, similar to what other authors report (in spite of having a much higher proportion of infectious etiologies, ranging from 55% to 79%). Another important aspect is that thanks to the ^18^F-FDG PET-CT results in cases in which infection was ruled out, patients were able to resume the chemotherapy treatment necessary for their underlying hematological disease, similar to other works [21].

Another benefit that ^18^F-FDG-PET-CT provides is its potential to detect dissemination of infection, especially in cases of IFI. Several articles state that ^18^F-FDG-PET-CT may have greater sensitivity than conventional tests to detect dissemination and occult lesions in the context of IFI [1,4,22]. In the analysis of our data, ^18^F-FDG-PET-CT was key in the diagnosis of disseminated fungal disease in one of the patients, which led to changes in therapeutic management.

Limitations to stating the role of ^18^F-FDG-PET-CT in febrile neutropenia workup are access and cost.

The systematic search retrieved several very heterogeneous articles that intended to evaluate the benefits of ^18^F-FDG-PET-CT in FN. The different methodologies, the lack of direct comparison between the techniques, and the different populations of patients studied precluded the performance of a metanalysis. With the exception of the only randomized controlled trial, the quality of the retrieved studies was poor. The lack of randomization together with the impossibility of masking increases the risk of bias. In this systematic review we analyze the results of these studies and point out knowledge gaps and unanswered questions.

First, although some of them are prospective studies, all are single-center studies that include only a small number of patients. The study by Mahfouz T et al. [18], even if it analyzes a large sample, is a retrospective study that only includes patients with MM, with a small proportion of FN. The studies by Koh et al. [17] and Guy et al. [19] were performed at the same hospital during the same period and might thus include in part the same patients.

We consider that the most relevant limitation of the majority of the articles is that they do not compare conventional tests and ^18^F-FDG-PET-CT to better discern what value the ^18^F-FDG-PET-CT adds in patients with FN, and among those that do (only five small studies) [14,15,17,19,20], not all perform both techniques on the same patient [17]. In this sense, the clinical trial by Douglas A. et al. [3] is a significant contribution in this area, but similar to others, its main limitation is not performing both tests in the same patient. As aforementioned, differences in underlying baseline characteristics or even in the cause of fever are difficult to address with this design and could explain, at least to some degree, the differences observed in the yields of the techniques that are being evaluated. When evaluating a diagnostic test, we believe it should be compared with other tests performed on the same patient.

Moreover, among those that compared ^18^F-FDG-PET-CT with conventional imaging, only one provided individual patient data [19]. This fact, in addition to the aforementioned methodologic heterogeneity, made it impracticable to perform a metanalysis.

In spite of these limitations, according to the results of the aforementioned articles that altogether include a total of 344 cases of FN, ^18^F-FDG-PET-CT seems to provide relevant information for the management of FN in a high proportion of cases.

In several of the studies the information provided by ^18^F-FDG PET-CT is especially relevant in the case of difficult to diagnose infections such as IFI [1,3,15,17,20] or parasitic infections. In addition to helping diagnose IFI and unveil dissemination, it also seems to be more useful to monitor the response to treatment than CT alone [23] since CT in some cases continues to show radiological lesions corresponding to scar tissue that do not show pathological uptake in ^18^F-FDG PET-CT, allowing the ending of antifungal treatment [1].

Of even greater importance is ^18^F-FDG PET-CT’s contribution to optimizing antimicrobial use in FN. Due to the high negative predictive value of ^18^F-FDG PET-CT, it allows reducing the use of broad-spectrum antimicrobials, favoring in many cases de-escalation and even discontinuation of empirical treatment, mainly of antifungals [1,3,17].

The clinical trial by Douglas et al. provides relevant information about the safety of basing clinical decisions on ^18^F-FDG PET-CT results. However, a formal cost-effectiveness analysis is pending to justify better access to ^18^F-FDG PET-CT in FN high-risk patients [3].

Clinicians experienced with the use of ^18^F-FDG PET-CT for the study of infection favor its use for prolonged FN and for an IFI diagnosis, according to a survey carried out by the Australian group. In particular, physicians who treat onco-hematological patients are likely to use ^18^F-FDG PET-CT in patients with FN to optimize the diagnosis and therapeutic management [24].

Many unanswered questions remain. In many cases, ^18^F-FDG PET-CT is considered when fever persists despite empirical antimicrobial treatments. But how long should we wait before performing ^18^F-FDG PET-CT? When will it perform better during the course of FN? Is there a basic workup that should be performed before considering ^18^F-FDG PET-CT? Should this basic set of studies include conventional imaging, or should ^18^F-FDG PET-CT replace them? Is there a particular subset of patients who would benefit more from ^18^F-FDG PET-CT? More studies with an adequate design are needed to clarify these points. We believe that a large multicentric prospective study that selects a well-categorized and homogeneous population of high-risk FN patients, using a protocolized workup for FN that includes both conventional imaging tests and ^18^F-FDG PET-CT performed on the same patient during a short pre-established time window, would be an appropriate model to clarify the role of ^18^F-FDG PET-CT and thus a diagnostic protocol that could include ^18^F-FDG-PET-CT.

## Figures and Tables

**Figure 1 microorganisms-12-00307-f001:**
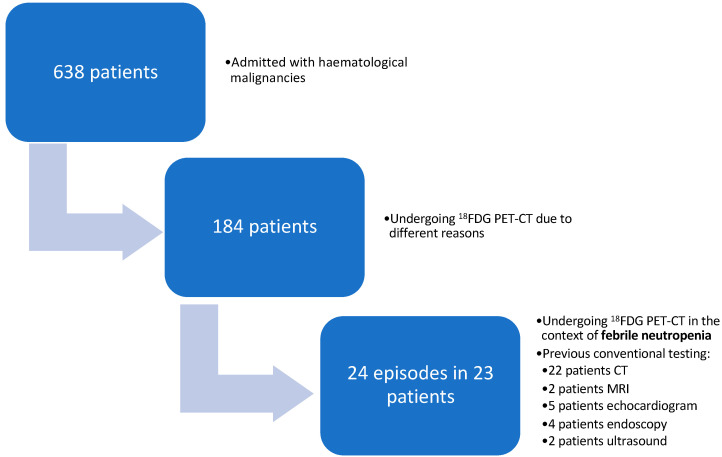
Patient selection.

**Figure 2 microorganisms-12-00307-f002:**
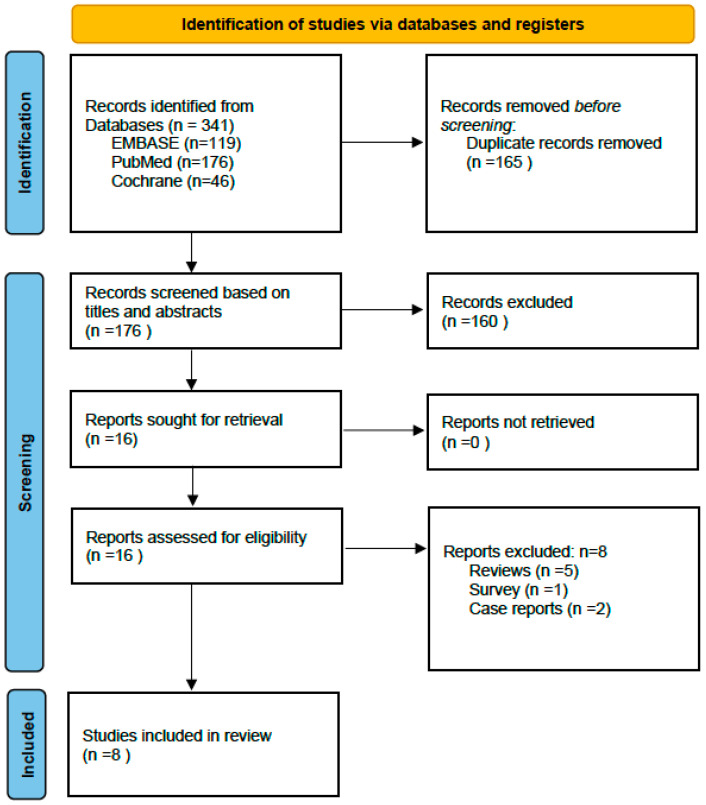
Systematic review.

**Table 1 microorganisms-12-00307-t001:** Research question in PICOS framework.

**P** (population): Patients of all ages with febrile neutropenia in the setting of oncohematologic malignancy
**I** (intervention): ^18^F-FDG PET-CT in the setting of febrile neutropenia management
**C** (comparison): Conventional diagnostic tests used for febrile neutropenia workup
**O** (outcome): ^18^F-FDG PET-CT **added value** in diagnosing the cause of fever in patients with febrile neutropenia: * Final diagnosis: infection, alternative non-infectious causes of fever ruling out infection * Directed targeted diagnostic tests * Antimicrobial rationalization (discontinuation, de-escalation, or escalation of antimicrobials) * Resumption of therapy of the underlying disease
**S** (study design): RCTs; prospective or retrospective cohort studies, case control studies, case reports yield

**Table 2 microorganisms-12-00307-t002:** Characteristics of FN episodes from our hospital.

Characteristic		Number (%)
**Number of patients**		23
**Number of FN episodes**		24
**Sex**		
	Male	13 (57%)
	Female	10 (43%)
**Age (years)** (median (IQR)/mean)		59 (47.5–74.5)/58.6
**Underlying non-oncological/haematological disease**		
	Coronary artery disease	0
	Heart failure	0
	Peripheral arterial disease	0
	Stroke	0
	Dementia	0
	Hemiplegia	0
	COPD	1 (4%)
	Diabetes	4 (17%)
	Diabetes with target organ involvement	0
	Moderate–severe kidney disease	2 (8%)
	Mild–moderate liver disease	0
	Severe liver disease	0
	Ulcer disease	0
	Connective tissue disease	0
	HIV-AIDS	0
**Onco-hematological disease**		
	Acute myeloid leukemia	12 (50%)
	Acute lymphoblastic leukemia	4 (17%)
	Multiple myeloma	3 (12%)
	Myelodysplastic syndrome	3 (12%)
	Others (NK immunodeficiency)	1 (4%)
	Lymphoma	0
**Stem cell transplantation**		4 (17%)
	Allogenic SCT	1 (4%)
	HLA-haploidentical SCT	2 (8%)
	Autologous SCT	1 (4%)
	Umbilical cord SCT	0
	GVHD post-SCT	2 (8%)
**Recent surgery (<3 months)**		3 (12%)
**Neutrophil counts on the day of ^18^F-FDG PET-CT**		
	<100 /mcL	12 (50%)
	100–500/mcL	10 (42%)
	500–1500/mcL	2 (8%)
**Fever**		
	Persistent low-grade fever (>72 h)	8 (33%)
	Persistent fever (>72 h)	14 (58%)
	Relapsing fever	1 (4%)
	Persistent–recurrent fever	1 (4%)
**Median (IQR)/mean days of neutropenia before ^18^F-FDG PET-CT**		13.5 (3–74)/19.3
**Median (IQR)/mean days of fever before ^18^F-FDG PET-CT**		13 (3–28)/13.7
**Median(IQR)/mean days from conventional test to ^18^F-FDG PET-CT**		13 (1–29)/12.9
**Median(IQR)/mean duration of antibiotic therapy until ^18^F-FDG PET-CT**		13.5 (3–30)/14
**Sepsis prior to ^18^F-FDG PET-CT (3 days)**		2 (8%)
**Antimicrobial prophylaxis**		
	Levofloxacin	15 (62%)
	Azithromycin	2 (8%)
	Cotrimoxazole	5 (21%)
	Acyclovir	11 (46%)
	Letermovir	0
	Entecavir	1 (4%)
	Fluconazole	1 (4%)
	Posaconazole	15 (62%)
	Pentamidine	1 (4%)
	Valganciclovir	1 (4%)
**Empiric treatment pre-^18^F-FDG PET-CT**		
	Cefepime	5 (21%)
	Piperacillin/tazobactam	9 (37%)
	Meropenem	13 (54%)
	Vancomicyn	4 (17%)
	Teicoplanin	11 (46%)
	Echinocandin	0 (0%)
	Voriconazole/isavuconazole	1 (4%)
	Ambisome	3 (12%)
	Others	3 (12%)
**Reference test ^18^F-FDG PET-CT**		
	Body CT	16 (67%)
	Chest CT	4 (17%)
	Abdominal CT	3 (12%)
	Sinus CT	2 (8%)
	MRI	2 (8%): hepatic 1; spinal 1.
	Echocardiogram (TTE)	3 (12%)
	Echocardiogram (TEE)	2 (8%)
	Endoscopy	2 (8%)
	Bronchoscopy	2 (8%)
	Abdominal ultrasound	2 (8%)
**Location of the ^18^F-FDG PET-CT uptake**		
	Lung	5 (21%)
	Brain	0
	Skin	2 (8%)
	Muscle	0
	Visceral intra-abdominal	9 (37%)
	Non-visceral intra-abdominal	1 (4%)
	Ocular	0
	Kidney	0
	Endocarditis	0
	Endovascular/cardiac devices	1 (4%)
	Bone marrow	5 (21%)
	Others	5 (21%)
**Type of uptake**		
	Focal	3 (12%)
	Multifocal	17 (71%)
	No uptake	4 (17%)
**Studies induced by ^18^F-FDG PET-CT result**		14 (58%)
	CT	2 (8%)
	MRI	0
	Echocardiogram	0
	Ultrasound	3 (12%)
	Endoscopy	3 (12%)
	Bronchoscopy	1 (4%)
	Biopsy/fine needle aspiration	4 (17%)
	Other	1 (4%)
**Source control of infection**		
	Catheter removal	1 (4%)
	Drain	0
	Surgery	1 (4%)
	Others	0
**Antimicrobial treatment modifications**		
Spectrum change (total)		16 (67%)
	Reduced spectrum	2 (8%)
	Extended spectrum	1 (4%)
	Started new treatment	3 (12%)
	Antibiotic discontinuation	10 (42%)
**Final diagnosis of infection**		
	Bacterial	2 (8%)
	Mycobacterial	0
	Fungal	2 (8%)
	Viral	0
	Parasitic	1 (4%)
	Clinical diagnosis without microbiological isolation	1 (4%)
**Site of infection**		
	Catheter	1 (4%)
	Cardiac	0
	Pulmonary	1 (4%)
	Urinary/genital	0
	Hepatic/biliary/splenic	2 (8%)
	Intestinal/perianal/oral	0
	Sinusitis	1 (4%)
	Skin and soft tissues	0
	Surgical site infection	0
**Non-infectious ethiology of fever**		
	Engraftment syndrome	3 (12%)
	Underlying hematological disease	11 (46%)
	Inflammatory	1 (4%)
	Gastric graft-versus-recipient disease	2 (8%)
	Febrile neutropenia of unknown origin	3 (12%)
**Added value of ^18^F-FDG PET-CT**		
	Diagnosis of new site of infection	4 (17%)
	Infection exclusion	16 (67%)
	Antibiotic modification	16 (67%)
	Induce other tests	10 (42%)
	Allows starting chemotherapy or immunomodulators	4 (17%)
**Discharge status (23 patients)**		
	Alive	14 (61%)
	Death	9 (39%)

**Table 3 microorganisms-12-00307-t003:** Evaluation of conventional imaging and added value of ^18^F-FDG PET-CT in 24 patients with febrile neutropenia.

Reference Conventional Imaging	Conventional Imaging Result	^18^F-FDG PET-CT Result	^18^F-FDG PET-CT vs. Conventional		Antimicrobial Therapy Modification	Induces New Diagnostic Tests	Final Diagnosis	Added Value
			Occult Lesions	Dissemination				
Full-body CT scan (11 October 2019)	CT: small intestine graft-versus-recipient disease	PET-CT (29 October 2019): Pathological uptake at gastroesophageal junction with gastric extension. Gallbladder uptake.	PET > CT PET scans show gastric uptake localized gastroesophageal junction + gallbladder	No	No	Ultrasound + gastroscopy with biopsies	Gastric graft-versus-recipient disease grade 4 and cholecystitis	Yes Occult lesion (colecistitis)
Chest-abdomen CT scan (17 April 2020) Liver MRI (18 April 2020)	CT: Decreased bilateral pulmonary nodules. MRI: Hepatic iron overload.	PET-CT (6 May 2020): Bone uptake secondary to MM (underlying disease) + uptake in the right thyroid lobe	PET > CT: Thyroid lobe was not visible on CT	No	Withdrawal of antibiotic treatment	Thyroid ultrasound	Tumor-related fever (multiple myeloma progression). Subclinical hyperthyroidism, BMN with right dominant thyroid nodule, TIRADS 3.	Yes Occult lesion Rule out infection Unveils tumoral etiology
Facial-sinus CT (30 April 2019) Abdominal CT scan (16 May 2019)	Sinus CT: Periodontal disease. No osteomyelitis. Abdominal CT: No urgent pathology.	PET-CT (17 May 2019): Retroperitoneal soft tissue uptake. Left pulmonary nodule without pathological uptake.	PET > CT Retroperitoneal uptake	No	No	Endoscopic ultrasound with biopsy of retroperitoneal lesions	Tumor-related fever (refractory AML). Disease progression with hepatic infiltration.	Yes Occult lesion Rule out infection Unveils tumoral etiology
Full-body CT scan (30 November 2020)	CT scan: Three splenic SOLs smaller than 10 mm, suggestive of infection	PET CT (15 December 2020): Splenic lesions without increased uptake	PET > CT Rules out infectious origin	No	Withdrawal of antibiotic treatment	Control abdominal ultrasound of splenic lesions	Tumor fever (acute biphenotypic leukemia, second relapse 3 m after second allogeneic transplant)	Yes Rule out infection Antibiotic discontinuation Induces initiation of hematological treatment
Full-body CT scan (13 April 2022)	Body CT: Decrease in bilateral pulmonary nodules consistent with IFI. Appearance of peribronchovascular GGO in LSD, suggestive of infection. Splenomegaly of 16 cm.	PET-CT (3 May 2022): Bilateral pulmonary nodules consistent with IFI. Splenomegaly without pathological uptake. Thyroid uptake.	PET = CT Confirm infection	No	Initiates amphotericin B	Bronchoscopy	Possible pulmonary IFI	Yes Antifungal started based on ^18^FDG PET-CT results Diagnosis of infectious source
Transesophageal echocardiogram (14 October 2019) Full-body CT scan (1 October 2019)	TEE: No signs of endocarditis. CT: Bilateral pleural effusion. Mixed patchy lung consolidations, pulmonary edema vs. infectious/inflammatory.	PET-CT (30 October 2019): Bilateral pulmonary consolidations with pathological uptake, especially those located in LSI, without clear superinfection, possible pulmonary edema. No cardiac uptake.	PET = CT Rules out lung infection and endocarditis	No	No	No	Bacteremia caused by *S. haemolyticus* in relation to central catheter, without local complications or endocarditis	No
Full-body CT scan (17 October 2022)	CT: Nonspecific pulmonary consolidations. Left hydropneumothorax.	PET-CT (8 November 2022): Uncomplicated left pleural effusion. Progression of hematologic disease.	PET > TC Rules out infection	No	No	No	Tumor-related fever. Graft failure. Relapse of hematologic disease (AML).	Yes Rules out infection Unveils tumoral etiology Initiates chemotherapy for relapse after HCST
Colonoscopy and panendoscopy (19 July 2018) No recent CT scan	Colonoscopy and panendoscopy with colitis and ileitis. Biopsies with GVHD.	PET-TC (7 August 2018): Diffuse intestinal uptake	Not applicable	No	No	No	Cutaneous and intestinal GVHD	Yes Rules out infection Initiates immunosuppression treatment for GVHD
Full-body CT scan (14 October 2019)	CT: No pathologic findings	PET-CT (19 November 2019): Diffuse bone marrow uptake. No pathological uptake.	PET > CT Rules out infection	No	De-escalation of antibiotic therapy	No	Tumor-related fever origin (newly diagnosed AML, refractory disease)	Yes Rule out infection De-escalate antibiotics
Chest CT (15 December 2018) Bronchoscopy (21 December 2018) Full-body CT scan (29 December 2018)	Chest CT: Patchy lesions of probable infectious etiology. BAL: No microbiological isolates. CT body: Bilateral peribronchovascular pulmonary micronodular involvement of the bronchiolitis type.	PET-CT (2 January 2019): Mandibular uptake and bilateral laterocervical lymphadenopathy in keeping with underlying process. Infectious pulmonary findings in resolution.	PET-CT > CT. Rule out hidden infection. Confirms improving previous lung infection.	No	Withdrawal of antibiotic therapy	No	Tumor-related fever (refractory AML)	Yes Confirms good response De-escalate antibiotics Unveils tumoral etiology
Full-body CT scan (20 December 2015)	CT body: radiological improvement of pulmonary infiltrates and hepatic SOLs	PET-CT (21 December 2015): Uptake of multiple hepatic SOL and abdominal lymphadenopathy	PET-CT > CT Rule out infectious complications	No	Withdrawal of antibiotic therapy	No	Previous diagnosis of systemic infection by atypical mycobacteria 10/2015. He was readmitted due to fever without a source. Diagnosis: fever of inflammatory origin (in relation to corticosteroid decrease).	Yes Discontinue antibiotics assuming inflammatory cause Rule out infection Increased corticosteroid doses
Chest CT (23 June 2022) Abdomen CT (29 June 2022)	Chest CT: Isolated centrolobular opacities in LII, probably inflammatory in nature. Abdomen CT: No findings.	PET-CT (30 June 2022): No pathological uptakes	PET-CT > CT Rule out infection	No	Withdrawal of antibiotic therapy	No	Tumor-related fever (AML progression)	Yes Rule out infection Antibiotic discontinuation
Full-body CT scan (29 June 2020)	CT: Normal chest. Renal angiomyolipomas. Left adrenal adenoma.	PET-CT (13 July 2020): Uptake in the left upper mola, inflammatory. No other pathological uptakes.	PET-CT > CT Rule out infection	No	De-escalation of antibiotic therapy	No	Fever in relation to neutrophil recovery	Yes Rule out infection Antibiotic de-escalation
Transesophageal echocardiogram (14 May 2021) Full-body CT scan (19 May 2021) Left upper limb Doppler ultrasound (21 May 2023)	TEE: No endocarditis. CT: Typhlitis. No pulmonary involvement. Doppler ultrasound: Postphlebitic axillary vein changes in relation to previous catheter.	PET-CT (28 May 2021): Active process in the nostril, of probable infectious origin. Cecum uptake in relation to typhlitis. Focal colon-sigma uptake of inflammatory vs. tumor etiology.	PET-CT > TC. Unveils more occult lesions (nasal, larger colonic disease) than CT	No	Increase in dose of amphotericin B and voriconazole is associated. Surgical debridement.	Colonoscopy: Colitis. Sinus CT: No complications. Periodontal disease. Evaluation by otolaryngologist: Biopsy. Positive culture for Fusarium.	Invasive fusariosis. Persistent bacteremia due to *E. faecium*, in relation to CVC without distant complications on PET. AML refractory to chemotherapy.	Yes Diagnosis of infectious source
Full-body CT scan (13 July 2021) Spinal MRI (15 July 2021)	CT body: Indeterminate scattered bone lesions. MR: Extensive changes in intraosseous marrow in relation to rapid bone loss or remodeling from treatments. Two areas of focal hypointensity, sclerose on CT, indeterminate, not possible to specify aggressiveness.	PET (20 July 2021): Pathological and diffuse uptake in bone marrow and lymph nodes at multiple levels in relation to underlying disease	PET-CT > CT Rule out infection	No	Withdrawal of antibiotic therapy	No	Tumor-related fever (ALL relapse)	Yes Rule out infection Antibiotic discontinuation
Full-body CT scan (12 April 2017)	CT: Pulmonary edema. Mild ascites. Mild pleural effusion.	PET-CT (19 April 2017): No pathological uptakes	PET-CT > CT Rule out infection	No	Withdrawal of antibiotic therapy	No	Tumor-related fever (progressive myelodysplastic syndrome)	Yes Rule out infection Antibiotic discontinuation
Transthoracic echocardiogram (19 May 2017) Abdominal ultrasound (20 May 2017)	TTE: No relevant findings. US: Hepatomegaly. No other findings.	PET-CT (1 June 2017): Uptake in front of both psoas, more on the right side, which translates into an active process (tumor vs. infectious/inflammatory)	PET > ultrasound	No	No	Abdominal CT scan (21 June 2017): No evidence of biopsy-eligible lesions	Febrile neutropenia of unknown origin	No
Full-body CT scan (8 February 2022) Bronchoscopy (9 February 2022)	CT: Small peripheral pulmonary infiltrates in RUL and RI	PET-CT (18 February 2022): Multiple diffuse pathological uptakes in subcutaneous tissue. Intense uptake in the subcutaneous tract of the right supraclavicular CVC without reaching vascular territory. Pulmonary infiltrates in RUL and right base, without pathological uptake.	PET-CT > CT Detects subcutaneous pathological uptake not observed on CT scan	No	Removal of CVC. Withdrawal of antibiotic therapy.	Skin biopsy: neutrophilic lobular panniculitis	Febrile neutropenia of unknown origin (probable inflammatory vs. tumor origin due to refractory AML)	Yes Rule out infection Antibiotic discontinuation Catheter removal
Chest CT (16 December 2021) Transthoracic echocardiogram (17 December 2021)	CT: Mild bilateral acinar opacities (edema vs. infectious-inflammatory). TTE: No findings.	PET-CT (18 December 2021): Very mild diffuse pulmonary uptake of dubious significance	PET-CT > CT Rule out infection	No	Withdrawal of antibiotic therapy	No	Fever in relation to engraftment syndrome	Yes Rule out infection Antibiotic discontinuation
Full-body CT scan (18 October 2021)	CT: Splenomegaly. Hepatomegaly with simple cysts. Bilateral adrenal thickening. Chronic bronchopulmonary disease.	PET-CT (24 October 2021): Splenomegaly with two foci with pathological uptake suggestive of splenic infarctions	PET-CT > CT Rule out infection	No	Withdrawal of antibiotic therapy	No	Tumor-related fever (AML)	Yes Rule out infection Antibiotic discontinuation
Full-body CT scan (10 February 2022)	CT: Mild pericardial effusion. Mild bilateral pleural effusion. Mild hepatosplenomegaly.	PET-CT (15 February 2022): Bone uptake suggestive of malignancy. Pericardial effusion without pathological uptake. Focal uptake in the left colon showing mild inflammatory origin associated with diverticulum.	PET-CT > CT Rule out infection	No	Withdrawal of antibiotic therapy	No	Febrile neutropenia of unknown origin (probable inflammatory vs. tumor origin due to refractory acute myeloid leukemia)	Yes Rule out infection Antibiotic discontinuation
Sinus CT (25 May 2022) Full-body CT scan (26 May 2022) Transthoracic echocardiogram (10 June 2022)	Sinus CT: Periorbital and soft tissue edema of the bilateral supratemporal fossa. CT body: Discrete continuous concentric parietal thickening of the colon suggestive of nonspecific colitis. ETT: No findings.	PET-CT (7 June 2022): Diffuse pancreatic uptake suggestive of inflammation. Pulmonary edema.	PET-CT > CT Rule out infection. Mild pancreatitis possible.	No	No	No	Tumor-related fever (newly diagnosed acute myeloid leukemia)	No
Chest CT (26 July 2022) Abdomen CT (27 July 2022)	CT thorax: Bibasal subsegmental atelectasis. CT scan of the abdomen: Slight parietal thickening of the colon suggestive of nonspecific colitis.	PET-CT (8 August 2022): No pathological uptake	PET-CT > CT Rule out infection	No	No	No	Febrile neutropenia with probable source mucositis vs. engraftment syndrome	No
Full-body CT scan (27 December 2022)	CT body: Homogeneous splenomegaly. Rest without significant findings.	PET-CT (1 January 2023): Splenomegaly with high-intensity diffuse uptake	PET-CT > CT New suspected source of infection	No	Initiation of amphotericin B	PCR leishmania in blood and bone marrow biopsy review	Visceral leishmaniasis	Yes Diagnosis of infectious source
Full-body CT scan (11 October 2019)	CT: small intestine graft-versus-recipient disease	PET-CT (29 October 2019): Pathological uptake at gastroesophageal junction with gastric extension. Gallbladder uptake.	PET > CT PET scans show gastric uptake localized gastroesophageal junction + gallbladder	No	No	Ultrasound + gastroscopy with biopsies	Gastric graft-versus-recipient disease grade 4 and cholecystitis	Yes Occult lesion (colecistitis)
Chest-abdomen CT scan (17 April 2020) Liver MRI (18 April 2020)	CT: Decreased bilateral pulmonary nodules. MRI: Hepatic iron overload.	PET-CT (6 May 2020): Bone uptake secondary to MM (underlying disease) + uptake in the right thyroid lobe	PET > CT Thyroid lobe was not visible on CT	No	Withdrawal of antibiotic treatment	Thyroid ultrasound	Tumor-related fever (multiple myeloma progression). Subclinical hyperthyroidism, BMN with right dominant thyroid nodule, TIRADS 3.	Yes Occult lesion Rule out infection Unveils tumoral etiology
Facial-sinus CT (30 April 2019) Abdominal CT scan (16 May 2019)	Sinus CT: Periodontal disease. No osteomyelitis. Abdominal CT: No urgent pathology.	PET-CT (17 May 2019): Retroperitoneal soft tissue uptake. Left pulmonary nodule without pathological uptake.	PET > CT Retroperitoneal uptake	No	No	Endoscopic ultrasound with biopsy of retroperitoneal lesions	Tumor-related fever (refractory AML). Disease progression with hepatic infiltration.	Yes Occult lesion Rule out infection Unveils tumoral etiology
Full-body CT scan (30 November 2020)	CT scan: Three splenic SOLs smaller than 10 mm, suggestive of infection.	PET CT (15 December 2020): Splenic lesions without increased uptake.	PET > CT Rules out infectious origin	No	Withdrawal of antibiotic treatment	Control abdominal ultrasound of splenic lesions	Tumor fever (acute biphenotypic leukemia, second relapse 3 m after second allogeneic transplant)	Yes Rule out infection Antibiotic discontinuation Induces initiation of hematological treatment
Full-body CT scan (13 April 2022)	Body CT: Decrease in bilateral pulmonary nodules consistent with IFI. Appearance of peribronchovascular GGO in LSD, suggestive of infection. Splenomegaly of 16 cm.	PET-CT (3 May 2022): Bilateral pulmonary nodules consistent with IFI. Splenomegaly without pathological uptake. Thyroid uptake.	PET = CT Confirm infection	No	Initiates amphotericin B	Bronchoscopy	Possible pulmonary IFI	Yes Antifungal started based on ^18^FDG PET-CT results Diagnosis of infectious source
Transesophageal echocardiogram (14 October 2019) Full-body CT scan (1 October 2019)	TEE: No signs of endocarditis.CT: Bilateral pleural effusion. Mixed patchy lung consolidations, pulmonary edema vs. infectious/inflammatory.	PET-CT (30 October 2019): Bilateral pulmonary consolidations with pathological uptake, especially those located in LSI, without clear superinfection, possible pulmonary edema. No cardiac uptake.	PET = CT Rules out lung infection and endocarditis	No	No	No	Bacteremia caused by *S. haemolyticus* in relation to central catheter, without local complications or endocarditis	No
Full-body CT scan (17 October 2022)	CT: Nonspecific pulmonary consolidations. Left hydropneumothorax.	PET-CT (8 November 2022): Uncomplicated left pleural effusion. Progression of hematologic disease.	PET > TC Rules out infection	No	No	No	Tumor-related fever. Graft failure. Relapse of hematologic disease (AML).	Yes Rules out infection Unveils tumoral etiology Initiates chemotherapy for relapse after HCST
Colonoscopy and panendoscopy (19 July 2018) No recent CT scan	Colonoscopy and panendoscopy with colitis and ileitis. Biopsies with GVHD.	PET-TC (7 August 2018): Diffuse intestinal uptake	Not applicable	No	No	No	Cutaneous and intestinal GVHD	Yes Rules out infection Initiates immunosuppression treatment for GVHD
Full-body CT scan (14 October 2019)	CT: No pathologic findings	PET-CT (19 November 2019): Diffuse bone marrow uptake. No pathological uptake.	PET > CT Rules out infection	No	De-escalation of antibiotic therapy	No	Tumor-related fever origin (newly diagnosed AML; refractory disease)	Yes Rule out infection De-escalate antibiotics
Chest CT (15 December 2018) Bronchoscopy (21 December 2018) Full-body CT scan (29 December 2018)	Chest CT: Patchy lesions of probable infectious etiology. BAL: No microbiological isolates. CT body: Bilateral peribronchovascular pulmonary micronodular involvement of the bronchiolitis type.	PET-CT (2 January 2019): Mandibular uptake and bilateral laterocervical lymphadenopathy in keeping with underlying process. Infectious pulmonary findings in resolution.	PET-CT > CT. Rule out hidden infection. Confirms improving previous lung infection.	No	Withdrawal of antibiotic therapy	No	Tumor-related fever (refractory AML)	Yes Confirms good response De-escalate antibiotics Unveils tumoral etiology
Full-body CT scan (20 December 2015)	CT body: Radiological improvement of pulmonary infiltrates and hepatic SOLs	PET-CT (21 December 2015): Uptake of multiple hepatic SOL and abdominal lymphadenopathy	PET-CT > CT Rule out infectious complications	No	Withdrawal of antibiotic therapy	No	Previous diagnosis of systemic infection by atypical mycobacteria 10/2015. He was readmitted due to fever without a source. Diagnosis: Fever of inflammatory origin (in relation to corticosteroid decrease).	Yes Discontinue antibiotics assuming inflammatory cause Rule out infection Increased corticosteroid doses
Chest CT (23 June 2022) Abdomen CT (29 June 2022)	Chest CT: Isolated centrolobular opacities in LII, probably inflammatory in nature. Abdomen CT: No findings.	PET-CT (30 June 2022): No pathological uptakes	PET-CT > CT Rule out infection	No	Withdrawal of antibiotic therapy	No	Tumor-related fever (AML progression)	Yes Rule out infection Antibiotic discontinuation
Full-body CT scan (29 June 2020)	CT: Normal chest. Renal angiomyolipomas. Left adrenal adenoma.	PET-CT (13 July 2020): Uptake in the left upper mola, inflammatory. No other pathological uptakes.	PET-CT > CT Rule out infection	No	De-escalation of antibiotic therapy	No	Fever in relation to neutrophil recovery	Yes Rule out infection Antibiotic de-escalation
Transesophageal echocardiogram (14 May 2021) Full-body CT scan (19 May 2021) Left upper limb Doppler ultrasound (21 May 2023)	TEE: No endocarditis. CT: Typhlitis. No pulmonary involvement. Doppler ultrasound: Postphlebitic axillary vein changes in relation to previous catheter.	PET-CT (28 May 2021): Active process in the nostril, of probable infectious origin. Cecum uptake in relation to typhlitis. Focal colon-sigma uptake of inflammatory vs. tumor etiology.	PET-CT > TC Unveils occult lesions (nasal, larger colonic disease than CT)	No	Increase in dose of amphotericin B and voriconazole is associated. Surgical debridement.	Colonoscopy: Colitis. Sinus CT: No complications. Periodontal disease. Evaluation by otolaryngologist: Biopsy. Positive culture for Fusarium.	Invasive Fusariosis. Persistent bacteremia due to *E. faecium*, in relation to CVC without distant complications on PET. AML refractory to chemotherapy.	Yes Diagnosis of infectious source
Full-body CT scan (13 July 2021) Spinal MRI (15 July 2021)	CT body: Indeterminate scattered bone lesions. MR: Extensive changes in intraosseous marrow in relation to rapid bone loss or remodeling from treatments. Two areas of focal hypointensity, sclerose on CT, indeterminate, not possible to specify aggressiveness.	PET (20 July 2021): Pathological and diffuse uptake in bone marrow and lymph nodes at multiple levels in relation to underlying disease	PET-CT > CT Rule out infection	No	Withdrawal of antibiotic therapy	No	Tumor-related fever (ALL relapse)	Yes Rule out infection Antibiotic discontinuation
Full-body CT scan (12 April 2017)	CT: Pulmonary edema. Mild ascites. Mild pleural effusion.	PET-CT (19 April 2017): No pathological uptakes	PET-CT > CT Rule out infection	No	Withdrawal of antibiotic therapy	No	Tumor-related fever (progressive myelodysplastic syndrome)	Yes Rule out infection Antibiotic discontinuation
Transthoracic echocardiogram (19 May 2017) Abdominal ultrasound (20 May 2017)	TTE: No relevant findings. US: Hepatomegaly. No other findings.	PET-CT (1 June 2017): Uptake in front of both psoas, more on the right side, which translates into an active process (tumor vs. infectious/inflammatory)	PET > ultrasound	No	No	Abdominal CT scan (21 June 2017): No evidence of biopsy-eligible lesions	Febrile neutropenia of unknown origin	No
Full-body CT scan (8 February 2022) Bronchoscopy (9 February 2022)	CT: Small peripheral pulmonary infiltrates in RUL and RI	PET-CT (18 February 2022): Multiple diffuse pathological uptakes in subcutaneous tissue. Intense uptake in the subcutaneous tract of the right supraclavicular CVC without reaching vascular territory. Pulmonary infiltrates in RUL and right base, without pathological uptake.	PET-CT > CT Detects subcutaneous pathological uptake not observed on CT scan	No	Removal of CVC. Withdrawal of antibiotic therapy.	Skin biopsy: Neutrophilic lobular panniculitis	Febrile neutropenia of unknown origin (probable inflammatory vs. tumor origin due to refractory AML)	Yes Rule out infection Antibiotic discontinuation Catheter removal
Chest CT (16 December 2021) Transthoracic echocardiogram (17 December 2021)	CT: Mild bilateral acinar opacities (edema vs. infectious-inflammatory). TTE: No findings.	PET-CT (18 December 2021): Very mild diffuse pulmonary uptake of dubious significance	PET-CT > CT Rule out infection	No	Withdrawal of antibiotic therapy	No	Fever in relation to engraftment syndrome	Yes Rul -out infection Antibiotic discontinuation
Full-body CT scan (18 October 2021)	CT: Splenomegaly. Hepatomegaly with simple cysts. Bilateral adrenal thickening. Chronic bronchopulmonary disease.	PET-CT (24 October 2021): Splenomegaly with two foci with pathological uptake suggestive of splenic infarctions	PET-CT > CT Rule out infection	No	Withdrawal of antibiotic therapy	No	Tumor-related fever (AML)	Yes Rule out infection Antibiotic discontinuation
Full-body CT scan (10 February 2022)	CT: Mild pericardial effusion. Mild bilateral pleural effusion. Mild hepatosplenomegaly.	PET-CT (15 February 2022): Bone uptake suggestive of malignancy. Pericardial effusion without pathological uptake. Focal uptake in the left colon showing mild inflammatory origin associated with diverticulum.	PET-CT > CT Rule out infection	No	Withdrawal of antibiotic therapy	No	Febrile neutropenia of unknown origin (probable inflammatory vs. tumor origin due to refractory acute myeloid leukemia)	Yes Rule out infection Antibiotic discontinuation
Sinus CT (25 May 2022) Full-body CT scan (26 May 2022) Transthoracic echocardiogram (10 June 2022)	Sinus CT: Periorbital and soft tissue edema of the bilateral supratemporal fossa. CT body: Discrete continuous concentric parietal thickening of the colon suggestive of nonspecific colitis. ETT: No findings.	PET-CT (7 June 2022): Diffuse pancreatic uptake suggestive of inflammation. Pulmonary edema.	PET-CT > CT Rule out infection. Mild pancreatitis possible.	No	No	No	Tumor-related fever (newly diagnosed acute myeloid leukemia)	No
Chest CT (26 July 2022) Abdomen CT (27 July 2022)	CT thorax: Bibasal subsegmental atelectasis. CT scan of the abdomen: Slight parietal thickening of the colon suggestive of nonspecific colitis.	PET-CT (8 August 2022): No pathological uptake	PET-CT > CT Rule out infection	No	No	No	Febrile neutropenia with probable source mucositis vs. engraftment syndrome	No
Full-body CT scan (27 December 2022)	CT body: Homogeneous splenomegaly. Rest without significant findings.	PET-CT (1 January 2023): Splenomegaly with high-intensity diffuse uptake	PET-CT > CT New suspected source of infection	No	Initiation of amphotericin B	PCR leishmania in blood and bone marrow biopsy review	Visceral leishmaniasis	Yes Diagnosis of infectious source

CT: computed tomography; TEE: transesophageal echocardiography; TTE: transthoracic echocardiography; SOL: space-occupying lesion; GGO: ground glass opacity; CVC: central venous catheter; RUL: right upper lobe.

## Data Availability

After publication, the data will be made available to others upon reasonable requests to the corresponding author. A proposal with a detailed description of study objectives and statistical analysis plan will be needed for evaluation of the reasonability of requests. It might also be required during the process of evaluation. Deidentified participant data will be provided after approval from the principal researchers of Hospital Universitario Puerta de Hierro (Majadahonda).

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
