# Peer review of "Usefulness of 18F-FDG PET-CT in the Management of Febrile Neutropenia: A Retrospective Cohort from a Tertiary University Hospital and a Systematic Review"

_microorganisms, 2024, doi:10.3390/microorganisms12020307_

Round 1
Reviewer 1 Report
Comments and Suggestions for Authors
This exciting topic regarding supportive care management of febrile neutropenic cancer patients with febrile neutropenia and using PET-CT imaging as a tool for confirmation or exclusion of infectious cause pathology.
- My comments to authors: PET-CT is usually done after persistent fever in most neutropenic patients, neutropenia most infectious lesions are undetected on CT imaging tools, and lesions are mostly seen after recovery. most of your patients did PET imaging after recovery of neutrophil count. Can you reply with the time difference between CT and PET the decision to do PET after standard imaging tools is usually late.
- What about SUV uptake of infectious lesions on PET-CT? any difference in uptake between bacterial and fungal pathogens in your small cohort.
- I recommend the authors to read this paper in (Clinical Features and Outcome of Hepatosplenic Fungal Infections in Children with Hematological Malignancies, Doi: 10.1111/myc.13002) as PET CT has been done in 10 pediatric cancer patients with hepatosplenic fungal infection with adding the concept of the role of PET-CT in management of febrile neutropenic cancer patients.
- PET-CT despite having a role in febrile neutropenia, you should highlight your in limitations regarding cost, non-availability in most centers.
Comments on the Quality of English Language
The quality English language is good
Author Response
Reviewer 1
Comments and Suggestions for Authors
This exciting topic regarding supportive care management of febrile neutropenic cancer patients with febrile neutropenia and using PET-CT imaging as a tool for confirmation or exclusion of infectious cause pathology.
Dear reviewer, thank you for your comments and contributions that we believe improve our manuscript.
- My comments to authors: PET-CT is usually done after persistent fever in most neutropenic patients, neutropenia most infectious lesions are undetected on CT imaging tools, and lesions are mostly seen after recovery. most of your patients did PET imaging after recovery of neutrophil count. Can you reply with the time difference between CT and PET the decision to do PET after standard imaging tools is usually late.
For this series, we selected only patients who were neutropenic at the moment of PET-CT. Only two patients had recovered neutrophils (<3 days before PET). Nevertheless, in table 3 you can see the day interval between tests. Median time from conventional tests to PET-CT was 12.9 days. This has been discussed in the manuscript.
- What about SUV uptake of infectious lesions on PET-CT? any difference in uptake between bacterial and fungal pathogens in your small cohort.
This is an interesting point, unfortunately, since this is a retrospective study, SUV data were not available for all the patients, so we could not perform this analysis. In future projects, we will consider analyzing this issue.
- I recommend the authors to read this paper in (Clinical Features and Outcome of Hepatosplenic Fungal Infections in Children with Hematological Malignancies, Doi: 10.1111/myc.13002) as PET CT has been done in 10 pediatric cancer patients with hepatosplenic fungal infection with adding the concept of the role of PET-CT in management of febrile neutropenic cancer patients.
We have read the recommended article and added a citation in the manuscript.
- PET-CT despite having a role in febrile neutropenia, you should highlight your in limitations regarding cost, non-availability in most centers.
Thank you for your comment. We have added a comment about those limitations in the manuscript discussion.
Reviewer 2 Report
Comments and Suggestions for Authors
Dear Authors,
congratulations on your valuable work. Please, find below some suggestions, that, in my opinion, could further improve the quality of your paper.
1. I suggest to remove the word "literature" from the title, it is obvious that you conducted a systematic review from literature.
2. Please improve the statistical analysis part. was the normality of the distribution of continuous variables studied? If yes, please indicate the method and specify how the variables are presented in the case of non-normal distribution. Please also specify the statistical tests used throughout the paper, the level of statistical significance set, the software used.
3. Regarding the systematic review, please specify whether it follows the recommendations of the PRISMA guidelines. Please improve the flow-chart (fig.2) as per the guidelines and add a check-list as recommended.
Author Response
Reviewer 2
Dear Authors,
congratulations on your valuable work. Please, find below some suggestions, that, in my opinion, could further improve the quality of your paper.
Dear reviewer, thank you for your comments and contributions.
- I suggest to remove the word "literature" from the title, it is obvious that you conducted a systematic review from literature.
Thank you for your contribution. We have corrected the title following your advice
- Please improve the statistical analysis part. was the normality of the distribution of continuous variables studied? If yes, please indicate the method and specify how the variables are presented in the case of non-normal distribution. Please also specify the statistical tests used throughout the paper, the level of statistical significance set, the software used.
This is a descriptive study, and, as such, there is no hypothesis contrast. It is not possible to provide a level of statistical significance, because no groups are compared. In this context, we consider that the distribution of continuous variables is not relevant. Measures of central tendency (mean and SD, median and IQR) were calculated with IBM SPSS Statistics 22. This has been added to the manuscript.
- Regarding the systematic review, please specify whether it follows the recommendations of the PRISMA guidelines. Please improve the flow-chart (fig.2) as per the guidelines and add a check-list as recommended.
We have followed the PRISMA guidelines for the systematic review. Thank you for pointing this out, we had forgotten to upload the PRISMA checklist. Now this information has been added to the manuscript and uploaded as supplementary material.